# Towards Optimal Communication Complexity in Distributed Non-Convex Optimization

**Kumar Kshitij Patel**[*]
TTIC
kkpatel@ttic.edu

**Lingxiao Wang**[*]
TTIC
lingxw@ttic.edu

**Blake Woodworth**
SIERRA, INRIA
blakewoodworth@gmail.com

**Brian Bullins**[†]
Purdue University
bbullins@purdue.edu

**Nati Srebro**
TTIC
nati@ttic.edu

## Abstract

We study the problem of distributed stochastic non-convex optimization with intermittent communication. We consider the full participation setting where $M$ machines work in parallel over $R$ communication rounds and the partial participation setting where $M$ machines are sampled independently every round from some meta-distribution over machines. We propose and analyze a new algorithm that improves existing methods by requiring fewer and lighter variance reduction operations. We also present lower bounds, showing our algorithm is either *optimal* or *almost optimal* in most settings. Numerical experiments demonstrate the superior performance of our algorithm.

## 1 Introduction

We consider the following distributed optimization problem with $M$ machines:

$$\min_{x \in \mathbb{R}^d} F(x) := \frac{1}{M} \sum_{m=1}^{M} F_m(x), \tag{1.1}$$

where $F_m$, which denotes the objective on machine $m$, is a non-convex function for all $m$, as is the average objective $F$. We want to solve this problem in the *intermittent communication* (IC) setting [1, 2] where the machines work in parallel and are allowed to make $K$ oracle calls between two communication rounds for $R$ consecutive rounds. The IC setting has been widely studied [3, 4, 5, 6, 7, 8, 9, 10, 11, 2, 12] over the past decade. Many recent works have focused on the problem with non-convex and heterogeneous objectives [13, 14, 15] which are common in cross-device federated learning (FL) [16, 17]. Towards this end, several algorithms [18, 19, 20, 21, 22, 23], all involving *local updates* (à la local-SGD [3, 16]), have been proposed and analyzed under assumptions bounding the heterogeneity of machines' objectives. Although these algorithms demonstrate promising empirical performances, it remains elusive whether these algorithms provably dominate the embarrassingly parallelizable alternatives, i.e., *mini-batch* variants of the optimal sequential algorithms [24, 25, 26] (a.k.a. centralized algorithms).

Until very recently, the situation was similar even in the simpler convex *homogeneous* setting where $F_m$'s are all identical and convex, and Woodworth et al. [2] showed that the optimal algorithm often does not require local updates at all. Even when $F_m$'s are not identical, for high levels of

---

[*]Equal contribution.     [†]This work was done while BB was a research assistant professor at TTIC.

heterogeneity, accelerated mini-batch SGD [27] is optimal [28]. Should we expect something similar in the non-convex setting? Or, can we prove that in some regime *local-update* algorithms improve over the naive centralized baselines?

| Method (Reference) (**Oracles used**) | Convergence Rate, i.e. $\mathbb{E}\|\nabla F(\widehat{x})\|^2 \preceq$ |
|---|---|
| **Full Participation Setting** | |
| SCAFFOLD[†], MB-SGD[†] [18] (**Stochastic**) | $\frac{\Delta L}{R} + \left(\frac{\sigma^2 \Delta L}{MKR}\right)^{1/2}$ |
| MB-STORM (Thm. C.2, [26]) (**Stochastic**) | $\frac{\Delta L}{R} + \frac{\sigma^2}{MKR} + \left(\frac{\sigma \Delta L}{MKR}\right)^{2/3}$ |
| Lower Bound (Centralized) (Theorem 2.1) | $\frac{\Delta L}{R} + \frac{\sigma^2}{MKR} + \left(\frac{\sigma \Delta L}{MKR}\right)^{2/3}$ |
| STEM [20] (**Stochastic**) | $(\Delta L + \sigma^2 + \zeta^2)\left(\frac{1}{R} + \frac{1}{(MKR)^{2/3}}\right)$ |
| BVR-L-SGD* [22] CE-LSGD (Thm. 3.1) (**Stochastic**) | $\frac{\Delta \tau}{R} + \frac{\Delta L}{\sqrt{K}R} + \frac{\sigma^2}{MKR} + \left(\frac{\sigma \Delta L}{MKR}\right)^{2/3}$ |
| CE-LGD (Thm. 3.1) (**Exact**) | $\frac{\Delta \tau}{R} + \frac{\Delta L}{KR}$ |
| Lower Bound (Theorem 3.2) | $\min\left\{\frac{\Delta \tau}{R}, \frac{\zeta^2}{R}\right\} + \frac{\Delta L}{KR} + \frac{\sigma^2}{MKR} + \left(\frac{\sigma \Delta L}{MKR}\right)^{2/3}$ |
| **Partial Participation Setting** | |
| MB-STORM (Thm. D.4) (**Stochastic**) | $\frac{\Delta L}{R} + \frac{\sigma^2}{MKR} + \left(\frac{\sigma \Delta L}{m\sqrt{K}R}\right)^{2/3} + \frac{\zeta^2}{mR} + \left(\frac{\zeta \Delta L}{mR}\right)^{2/3}$ |
| Lower Bound (Centralized) (Theorem D.2) | $\frac{\Delta L}{R} + \frac{\sigma^2}{mKR} + \left(\frac{\sigma \Delta L}{mKR}\right)^{2/3} + \frac{\zeta^2}{mR} + \left(\frac{\zeta \Delta L}{mR}\right)^{2/3}$ |
| MIMELITEMVR[21] (**Stochastic + Exact**) | $\frac{\Delta \tau}{R} + \frac{\Delta L}{KR} + \frac{\zeta^2 + \sigma^2}{R} + \left(\frac{(\zeta + \sigma)\Delta \tau}{R}\right)^{2/3}$ |
| MIMEMVR [21] (**Exact**) | $\frac{\Delta \tau}{R} + \frac{\Delta L}{KR} + \frac{\zeta^2}{mR} + \left(\frac{\zeta \Delta \tau}{\sqrt{m}R}\right)^{2/3}$ |
| CE-LSGD (Thm. 3.3) (**Stochastic**) | $\frac{\Delta \tau}{R} + \frac{\Delta L}{\sqrt{K}R} + \frac{\sigma^2}{mKR} + \left(\frac{\sigma \Delta L}{mKR}\right)^{2/3} + \left(\frac{\sigma \Delta \tau}{m\sqrt{K}R}\right)^{2/3} +$ $\frac{\zeta^2}{mR} + \left(\frac{\zeta \Delta \tau}{mR}\right)^{2/3} + \left(\frac{\zeta \Delta L}{m\sqrt{K}R}\right)^{2/3}$ |
| CE-LGD (Thm. 3.3) (**Exact**) | $\frac{\Delta \tau}{R} + \frac{\Delta L}{KR} + \frac{\zeta^2}{mR} + \left(\frac{\zeta \Delta \tau}{mR}\right)^{2/3}$ |
| Lower Bound (Thm. 3.4) | $\min\left\{\frac{\Delta \tau}{R}, \frac{\zeta^2}{R}\right\} + \frac{\Delta L}{KR} + \frac{\sigma^2}{mKR} + \left(\frac{\sigma \Delta L}{mKR}\right)^{2/3} + \frac{\zeta^2}{mR} +$ $\left(\frac{\zeta \Delta L}{mKR}\right)^{2/3}$ |

Table 1: Comparison of convergence rate for different algorithms in the intermittent communication setting. $\zeta$ and $\tau$ are the first and second-order heterogeneity (see Section 2) of the problem. Note that $\tau \le 2L$ can be much smaller than $L$. *See Section 3 for a detailed comparison with BVR-L-SGD. We expect the red and blue terms in the bounds to match by improving our bounds (c.f., Section 5). [†]The variance term is optimal as the algorithms' analyses don't assume mean squared smoothness.

In this paper, we start by noting that in the absence of any heterogeneity assumption (c.f., Section 2), centralized algorithms already have the best worst-case convergence guarantee. Thus, only when

the heterogeneity is low can the *local-update* algorithms potentially have an advantage. This was the motivation behind some of the recent works [18, 21, 22]. However, in the absence of any lower bound that explicitly depends on the heterogeneity parameter (such as in [15, 29]), it is not possible to definitively claim such an improvement. To alleviate this, we provide new communication lower bounds which explicitly depends on the heterogeneity parameter. In addition, we develop a novel algorithm which can take advantage of low heterogeneity and is (almost) optimal.

| Method (Reference) | Communication Complexity ($R$) | Oracle Complexity ($N$) |
|---|:---:|:---:|
| **Full Participation Setting** | | |
| SCAFFOLD[†], MB-SGD[†] [18] | $\frac{\Delta L}{\epsilon}$ | $\frac{\sigma^2 \Delta L}{\epsilon^2}$ |
| MB-STORM (Theorem C.2) [26] | $\frac{\Delta L}{\epsilon}$ | $\frac{\sigma \Delta L}{\epsilon^{3/2}}$ |
| Lower Bound (Centralized) (Theorem 2.1) | $\frac{\Delta L}{\epsilon}$ | $\frac{\sigma \Delta L}{\epsilon^{3/2}}$ |
| STEM [20] | $\frac{\Delta L + \sigma^2 + \zeta^2}{\epsilon}$ | $\frac{(\Delta L)^{3/2} + \sigma^3 + \zeta^3}{\epsilon^{3/2}}$ |
| BVR-LSGD* [22] CE-LSGD (Theorem 3.1) | $\frac{\Delta \tau}{\epsilon}$ | $\frac{\sigma \Delta L}{\epsilon^{3/2}}$ |
| Lower Bound (Theorem 3.2) | $\min\left\{\frac{\Delta \tau}{\epsilon}, \frac{\zeta^2}{\epsilon}\right\}$ | $\frac{\sigma \Delta L}{\epsilon^{3/2}}$ |
| **Partial Participation Setting** | | |
| MB-STORM (Theorem C.2) | $\frac{\zeta \Delta L}{m \epsilon^{3/2}}$ | $\frac{\sigma \Delta L}{\epsilon^{3/2}} \cdot \left(1 + \frac{\sigma}{\zeta}\right)$ |
| Lower Bound (Centralized) (Theorem 2.1) | $\frac{\zeta \Delta L}{m \epsilon^{3/2}}$ | $\frac{\sigma \Delta L}{\epsilon^{3/2}}$ |
| MIMEMVR [21] | $\frac{\zeta \Delta \tau}{m^{1/2} \epsilon^{3/2}}$ | **Uses Exact Oracles** |
| MIMELITEMVR [21] | $\frac{\zeta^2 + \sigma^2}{\epsilon} + \frac{(\zeta + \sigma)\Delta \tau}{\epsilon^{3/2}}$ | **Uses Exact Oracles** |
| CE-LSGD (Theorem 3.3) | $\frac{\zeta \Delta \tau}{m \epsilon^{3/2}}$ | $\frac{\zeta \Delta L}{\epsilon^{3/2}} \cdot \frac{L}{\tau} + \frac{\sigma \Delta L}{\epsilon^{3/2}} \cdot \left(1 + \frac{\sigma \tau}{\zeta L}\right)$ |
| Lower Bound (Theorem 3.4) | $\min\left\{\frac{\Delta \tau}{\epsilon}, \frac{\zeta^2}{\epsilon}\right\} + \frac{\zeta^2}{m \epsilon}$ | $\frac{\zeta \Delta L}{\epsilon^{3/2}} + \frac{\sigma \Delta L}{\epsilon^{3/2}}$ |

Table 2: Comparison of optimal communication and oracle complexity required by different algorithms to attain $\mathbb{E}\|\nabla F(\widehat{x})\|_2^2 \le \epsilon$. $\zeta$ and $\tau$ are the heterogeneity (see Section 2) of the problem. $\tau \le 2L$ and can be much smaller than $L$. The results suppress only numerical constants and assume that $\epsilon^{1/2} \preceq \min\{(\sigma/M) \cdot (\tau/L), \Delta L/\sigma, \Delta \tau/\zeta, \zeta/m\}$, i.e., $\epsilon$ is small enough. The first inequality ensures we are in the green regime described in Figure 1 and guarantees that $\Delta LM/\epsilon \preceq \sigma \Delta L/\epsilon^{3/2}$; the second inequality guarantees that $\sigma^2/\epsilon \preceq \sigma \Delta L/\epsilon^{3/2}$; the third inequality guarantees that $\zeta^2/m\epsilon \preceq \zeta \Delta \tau/m\epsilon^{3/2}$; and the fourth inequality guarantees that $\Delta L/\epsilon \le \zeta \Delta L/m\epsilon^{3/2}$. We expect the red, green, and blue terms in the upper and lower bounds to match by improving our bounds (c.f., Section 5). *Although BVR-L-SGD and CE-LSGD have the same fast convergence rate in the full participation setting, BVR-L-SGD requires each client to compute large batch gradients for many rounds of communications and is thus less communication efficient in practice (see discussion in Section 3). [†]Note that the oracle complexity is optimal for these algorithms, as they were analyzed under the bounded variance assumption (see Section 2).

We summarize the contributions of our work as follows:

- We provide novel communication complexity lower bounds, under the assumption that $F_m$'s have bounded first-order or second-order heterogeneity (see Section 2). We show that centralized algorithms [24, 25, 26] can never achieve this optimal communication complexity, and most of the existing *local-update* algorithms cannot attain it either.

- We develop a new algorithm **CE-LSGD** that we show to be **min-max optimal when equipped with exact gradient oracles** and near-optimal when provided with stochastic gradient oracles (c.f., Section 2). Our algorithm, like many other *local-update* algorithms, uses variance reduction techniques [24, 26] but requires both fewer and lighter "heavy-batch" operations compared to the existing methods (see discussion in Section 3).

- We also study the partial client participation setting, which is of particular interest in cross-device federated learning (FL) [17] where there is an extremely large number of clients. Not only does CE-LSGD improve over the best-known communication complexity, but it is the only algorithm that *doesn't require exact oracle* queries for variance reduction and still manages to be nearly optimal. Our analysis also provides a convergence guarantee for MB-STORM (a special case of CE-LSGD) in this setting that wasn't known before. Furthermore, if **endowed with exact oracles, CE-LGD is almost min-max optimal even in the partial participation setting**. Thus, our results demonstrate the optimality of local update methods, at least in some regimes. Even in simpler convex settings, we don't know of any local update method (exact or stochastic) known to be min-max optimal in the heterogeneous setting [30, 15]. We summarize our results and the comparison to important baselines in Tables 1 and 2.

- As an auxiliary contribution, we provide a variant of our algorithm which uses stochastic hessian vector product oracles and is thus useful for settings where only a single copy of the model can be stored on the edge device. We also empirically compare our method against centralized and *local-update* algorithms, demonstrating faster convergence and better communication efficiency.

**Notation.** We use $\mathcal{B}$ to denote the index set and $|\mathcal{B}|$ to denote its cardinality. For $x \in \mathbb{R}^d$, we use $\|x\|$ to denote its $\ell_2$-norm. For $A \in \mathbb{R}^{d \times d}$, $\|A\|$ denotes the operator norm. $[n]$ denotes the set $\{1, 2, \ldots, n\}$. We use $\approx, \preceq, \succeq$ to denote equality and inequality up to numerical constants.

## 2  Our Setting and the Centralized Baselines

In this section, we introduce some definitions and assumptions that will be used in our analysis. Our goal is to find an $\epsilon$-approximate stationary point of $F$, i.e., a point $x \in \mathbb{R}^d$ such that $\mathbb{E}[\|\nabla F(x)\|^2] \leq \epsilon$, where the expectation is w.r.t. any randomness in the choice of $x$. We consider client objectives in the class $\mathcal{F}(L)$ of differentiable and $L$-smooth functions, i.e., for all $G \in \mathcal{F}(L)$, $\|\nabla G(x) - \nabla G(y)\| \leq L \|x - y\|$. We also make assumptions that relate the functions of different clients to one another. These are typically known as assumptions on the *"heterogeneity"* of the problem, and we consider two classes of problems.

**Definition 1.** *Assume* $\{F_m \in \mathcal{F}(L)\}_{m=1}^M$ *are first-order* $\zeta$*-heterogeneous, i.e.,* $\sup_{x \in \mathbb{R}^d} \sum_{m=1}^M \|\nabla F_m(x) - \nabla F(x)\|^2/M \leq \zeta^2$. *And for all* $\Delta \geq 0$, $F(0) - \inf_{x \in \mathbb{R}^d} F(x) \leq \Delta$, *i.e., the average objective has bounded sub-optimality at zero. Then we say that* $\{F_m\}_{m \in [M]} \in \mathcal{F}_M^1(L, \Delta, \zeta)$.

**Definition 2.** *Assume twice-differentiable* $\{F_m \in \mathcal{F}(L)\}_{m=1}^M$ *are second-order* $\tau$*-heterogeneous, i.e.,* $\sup_{m \in [M], x \in \mathbb{R}^d} \|\nabla^2 F_m(x) - \nabla^2 F(x)\| \leq \tau$. *And for all* $\Delta \geq 0$, $F(0) - \inf_{x \in \mathbb{R}^d} F(x) \leq \Delta$, *i.e., the average objective has bounded sub-optimality at zero. Then we say that* $\{F_m\}_{m \in [M]} \in \mathcal{F}_M^2(L, \Delta, \tau)$.

We assume that each machine has access to the following multi-point oracle [31] [Section 5.3, 2].

**Definition 3.** *Given a function* $G \in \mathcal{F}(L, \Delta)$, $\mathcal{O}_G^{n, L, \sigma} : (\mathbb{R}^d)^n \times \mathcal{Z} \to (\mathbb{R})^n \times (\mathbb{R}^d)^n$ *is a multi-point stochastic first order oracle if for some distribution* $\mathcal{D}$ *on* $\mathcal{Z}$, *and for all* $x_1, \ldots, x_n \in \mathbb{R}^d$, *the oracle samples a random seed* $z \sim \mathcal{D}$ *and returns estimators* $\mathcal{O}_G^{n, L, \sigma}(x_1, \ldots, x_n, z) = (\{f(x_i; z)\}_{i \in [n]}, \{g(x_i; z)\}_{i \in [n]})$ *such that* $\forall i \in [n]$, $\mathbb{E}_{z \sim \mathcal{D}}(f(x_i; z), g(x_i; z)) = (G(x_i), \nabla G(x_i))$ *and* $\mathbb{E}_{z \sim \mathcal{D}} \|g(x_i; z) - \nabla G(x_i)\|^2 \leq \sigma^2$. *Furthermore, the unbiased gradients satisfy L-mean smoothness, i.e., for all* $x, y \in \mathbb{R}^d$, $\mathbb{E}_{z \sim \mathcal{D}}[\|g(x; z) - g(y; z)\|] \leq L \|x - y\|$.

As we mentioned before, we want to solve the problem in equation 1.1 in the the intermittent communication (IC) setting, i.e., $M$ machines work in parallel and are allowed to make $K$ oracle calls between two communication rounds for $R$ consecutive rounds (see [1, 2] for detailed definition). Therefore, we consider a generalization of zero-respecting algorithms denoted by $\mathcal{A}_{ZR}$ (see Appendix A) in the IC setting. This class captures various distributed optimization algorithms, including mini-batch SGD, accelerated mini-batch SGD, local SGD, and all the variance-reduction algorithms. Algorithms that are not distributed zero-respecting are those whose iterates have components in directions about which the algorithm has no information, meaning that, in some sense, it is just "wild guessing". We also denote the class of centralized algorithms in $\mathcal{A}_{ZR}$ by $\mathcal{A}_{ZR}^{cent}$ (see Appendix A). These algorithms query the oracles at the same point within each communication round and use the combined $MK$ oracle queries each round to get a *"mini-batch"* estimate of the gradient. Thus, the class $\mathcal{A}_{ZR}^{cent}$ includes algorithms such as mini-batch SGD, mini-batch SARAH [24] and mini-batch STORM [26], but doesn't include local-update algorithms in $\mathcal{A}_{ZR}$ such as local-SGD. Furthermore, these mini-batch algorithms can be naturally implemented in the IC setting.

We first present a lower bound result applicable to centralized algorithms.

**Theorem 2.1** (Centralized Lower Bound). *For all $\tau, \Delta, \zeta, \sigma \geq 0$, and $2L \geq \tau$, every algorithm $A \in \mathcal{A}_{ZR}^{cent}$ optimizing a problem in $\mathcal{F}_M^1(L, \Delta, \zeta) \cup \mathcal{F}_M^2(L, \Delta, \tau)$, with access to an oracle $\mathcal{O}_{F_m}^{2,L,\sigma}$ over $R \succeq 1$ communication rounds must output $x_R^A$ such that,*

$$\mathbb{E}\left[\left\|\nabla F(x_R^A)\right\|^2\right] \succeq \frac{\Delta L}{R} + \frac{\sigma^2}{MKR} + \left(\frac{\sigma \Delta L}{MKR}\right)^{2/3}.$$

The proof of this theorem follows the known oracle complexity lower bounds [32, 31] and is included in Appendix B. This theorem shows that, mini-batch SARAH/STORM which are centralized algorithms, already achieve the optimal communication and oracle complexity (see Table 1) for algorithms in $\mathcal{A}_{ZR}^{cent}$ optimizing problems in $\mathcal{F}_M^2(L, \Delta, \tau)$. In fact most existing non-centralized methods including FEDAVG[16], SCAFFOLD [18] and FEDPAGE [19] do not have any analysis showing improvement over the centralized baselines for problems in $\mathcal{F}_M^2(L, \Delta, \tau)$. These analyses do not improve with smaller heterogeneity $\tau$, even for convex optimization problems. At the same time, the lower bound result holds for all $\tau \leq 2L$, which highlights the limitation of the centralized baselines, showing they **can not** improve with lower heterogeneity. Certain existing *local-update* algorithms such as MIMEMVR [21] and BVR-L-SGD [22] can indeed improve upon centralized algorithms in the low heterogeneity regime. In the next section, we quantify this improvement and show that our algorithm strictly dominates the centralized baselines and almost matches our lower bound for algorithms in $\mathcal{A}_{ZR}$.

## 3 Our Algorithm and Min-max Optimality

In this section, we present our communication-efficient algorithm abbreviated CE-LSGD and illustrate it in Algorithm 1. Note that for $m \in [M]$, we use the notation $\nabla F_{m,\mathcal{B}^m}(x) := \sum_{l \in \mathcal{B}^m} g(x; z_l \sim \mathcal{D}_m)/|\mathcal{B}^m|$ to denote the stochastic mini-batch gradient obtained by querying $\mathcal{O}_{F_m}^{2,L,\sigma}$ for $|\mathcal{B}^m|$ many times.

At each iteration of Algorithm 1, we need **two rounds** of communication, i.e., two back and forth communications between the server and all clients. Our method uses the extra round of communication, i.e., line 4 to line 9, to update the variance-reduced gradient $v_r$ using the current and previous server models $x_r, x_{r-1}$, respectively. In the following discussion, we will use the iteration number $R$ and communication complexity of Algorithm 1 interchangeably.

At the core of our proposed method is the variance reduction term $v_r$ and the local gradient estimator $v_{r,k}^m$ (lines 9 and 15 in Algorithm 1). The construction of the local gradient estimator is motivated by the variance reduction technique of SARAH [24, 25]. Intuitively, the estimation error between the proposed local gradient estimator $v_{r,k}^m$ and the full gradient $\nabla F(w_{r+1,k}^m)$, i.e., $\mathbb{E}\|v_{r,k}^m - \nabla F(w_{r+1,k}^m)\|$, can be decomposed into two dominating terms: $\mathbb{E}\|v_r - \nabla F(x_r)\|^2$ and $\tau^2 K \sum_{k=1}^K \mathbb{E}\|w_{r+1,k}^m - w_{r+1,k-1}^m\|^2$. The first term is the estimation error between the variance reduction term $v_r$ and the full gradient $\nabla F(x_r)$. Since $v_r$ is updated based on the momentum-based

---

**Algorithm 1** Communication Efficient Local Stochastic Gradient Descent (CE-LSGD)

---

**input** Initialization $x_0$, iteration number $R$, step size $\eta$, parameters $b_0$, $b$, $T$ and $\beta \in [0, 1]$

1: Let $x_{-1} = x_0$
2: **for** $r = 0, 1, \ldots, R - 1$ **do**
3:     **if** $r = 0$ set $\rho = 1$, $Q = 1$, $B = b_0$ **else** set $\rho = \beta$, $Q = T$, $B = Q$
4:     **Communicate (send)** $(x_r, x_{r-1})$ to clients
5:     **on client** $m \in [M]$ **do**
6:         Sample $\mathcal{B}_r^m \sim \mathcal{D}_m^{\otimes B}$, get $\nabla F_{m, \mathcal{B}_r^m}(x_r), \nabla F_{m, \mathcal{B}_r^m}(x_{r-1})$, where $|\mathcal{B}_r^m| = B$
7:         **Communicate (rec)** $\left( \nabla F_{m, \mathcal{B}_r^m}(x_r), \nabla F_{m, \mathcal{B}_r^m}(x_{r-1}) \right)$ to the server
8:     **end on client**
9:     $v_r = \frac{1}{M} \sum_{m=1}^M \nabla F_{m, \mathcal{B}_r^m}(x_r) + (1 - \rho) \left( v_{r-1} - \frac{1}{M} \sum_{m=1}^M \nabla F_{m, \mathcal{B}_r^m}(x_{r-1}) \right)$
10:     **Communicate (send)** $(x_r, v_r)$ to client $\widetilde{m}_r$, where $\widetilde{m}_r \sim Unif([M])$
11:     **on client** $\widetilde{m}_r$ **do**
12:         $w_{r+1,1}^{\widetilde{m}_r} := w_{r+1,0}^{\widetilde{m}_r} := x_r, v_{r,0}^{\widetilde{m}_r} := v_r$
13:         **for** $k = 1, \ldots, Q$ **do**
14:             Sample $\mathcal{B}_{r,k}^{\widetilde{m}} \sim \mathcal{D}_{\widetilde{m}}^{\otimes b}$, get $\nabla F_{\widetilde{m}, \mathcal{B}_{r,k}^{\widetilde{m}}}(w_{r+1,k}^{\widetilde{m}_r}), \nabla F_{\widetilde{m}, \mathcal{B}_{r,k}^{\widetilde{m}}}(w_{r+1,k-1}^{\widetilde{m}_r})$, where $|\mathcal{B}_{r,k}^{\widetilde{m}}| = b$
15:             $v_{r,k}^{\widetilde{m}_r} = \nabla F_{\widetilde{m}, \mathcal{B}_{r,k}^{\widetilde{m}}}(w_{r+1,k}^{\widetilde{m}_r}) + v_{r,k-1}^{\widetilde{m}_r} - \nabla F_{\widetilde{m}, \mathcal{B}_{r,k}^{\widetilde{m}}}(w_{r+1,k-1}^{\widetilde{m}_r})$
16:             $w_{r+1,k+1}^{\widetilde{m}_r} = w_{r+1,k}^{\widetilde{m}_r} - \eta v_{r,k}^{\widetilde{m}_r}$
17:         **end for**
18:         **Communicate (rec)** $\left( w_{r+1,Q+1}^{\widetilde{m}_r} \right)$ to the server
19:     **end on client**
20:     Let $x_{r+1} = w_{r+1,Q+1}^{\widetilde{m}_r}$
21: **end for**
**output** Choose $\widetilde{x}$ uniformly from $\{ w_{r,k}^{\widetilde{m}_r} \}_{r \in [R], k \in [Q]}$

---

variance reduction technique [26], this estimation error is dominated by $L^2 \mathbb{E}\|x_r - x_{r-1}\|^2$ , which approaches zero as the algorithm converges. Similarly, the second term $\mathbb{E}\|w_{r+1,k}^m - w_{r+1,k-1}^m\|^2$ approaches zero as the algorithm converges and the $\tau$ factor controls the benefit we can obtain from small heterogeneity. Intuitively, we can make more local updates for smaller values of $\tau$, and the algorithm converges faster. Our method reduces to mini-batch STORM if we choose the number of local updates $Q$ to be one (see Appendix C.1).

As we mentioned before, we are considering the IC setting, and thus we want to implement Algorithm 1 in this setting. To this end, we can choose the input $T = K$ and $b = 1$ (see line 14) in Algorithm 1, and we present the convergence guarantees of our method in the IC setting in the following discussions. Next we present the convergence guarantee of CE-LSGD in the intermittent communication:

**Theorem 3.1.** *Suppose $\{F_m\}_{m \in [M]} \in \mathcal{F}_M^2(L, \Delta, \tau)$ for $L, \Delta, \tau \geq 0, \tau \leq 2L$ then,*

*(a) if each client $m \in [M]$ has a stochastic oracle $\mathcal{O}_{F_m}^{2,L,\sigma}$, and assuming $\frac{\Delta L}{R} \preceq \frac{\sigma^2}{\sqrt{MK}}$, then the output $\widetilde{x}$ of Algorithm 1 using $\beta = \max\left\{ \frac{1}{R}, \frac{(\Delta L)^{2/3}(MK)^{1/3}}{\sigma^{4/3}R^{2/3}} \right\}$, $b_0 = KR$, and $\eta = \min\left\{ \frac{1}{L}, \frac{1}{K\tau}, \frac{(\beta M)^{1/2}}{LK^{1/2}} \right\}$ satisfies*

$$\mathbb{E}\|\nabla F(\widetilde{x})\|^2 \preceq \frac{\Delta \tau}{R} + \frac{\Delta L}{\sqrt{K}R} + \frac{\sigma^2}{MKR} + \left( \frac{\sigma \Delta L}{MKR} \right)^{2/3};$$

*(b) if each client $m \in [M]$ has a deterministic oracle $\mathcal{O}_{F_m}^{2,L,0}$, then the output $\widetilde{x}$ of Algorithm 1 using $\beta = 1$ and $\eta = \min\left\{ \frac{1}{L}, \frac{1}{K\tau} \right\}$ satisfies*

$$\mathbb{E}\|\nabla F(\widetilde{x})\|^2 \preceq \frac{\Delta \tau}{R} + \frac{\Delta L}{KR}.$$

In Appendix C, we derive this result by carefully tuning $\beta, b_0$. We show that the convergence rate attained by our algorithm is *almost optimal* by proving the following lower bound result.

**Theorem 3.2.** *For all $L, \sigma, \tau, \Delta, \zeta \geq 0$, $\tau \leq 2L$, $\zeta \leq \sqrt{\Delta L}$, every algorithm $A \in \mathcal{A}_{zr}$, optimizing a problem in $\mathcal{F}_M^1(L, \Delta, \zeta) \cup \mathcal{F}_M^2(L, \Delta, \tau)$ with $K > 0$ intermittent accesses to two-point first-order oracles $\{\mathcal{O}_{F_m}^{2,L,\sigma}\}_{m \in [M]}$ on all the machines, outputs $x_R^A$ after $R \succeq 1$ rounds such that*

$$\mathbb{E}\left[\left\|\nabla F(x_R^A)\right\|^2\right] \succeq \min\left\{\frac{\zeta^2}{R}, \frac{\Delta\tau}{R}\right\} + \frac{\Delta L}{KR} + \frac{\sigma^2}{MKR} + \left(\frac{\sigma\Delta L}{MKR}\right)^{2/3}.$$

We can make two observations by comparing the upper and lower bounds for problems in $\mathcal{F}_M^2(L, \Delta, \tau)$. First, in the deterministic setting ($\sigma = 0$), our upper bound matches the lower bound; **hence CE-LGD is min-max optimal**. Thus, our result improves over all the existing results in this setting, including MIMEMVR [21]. Second, in the stochastic setting ($\sigma > 0$), our algorithm's upper bound is optimal except for the second term in Theorem 3.1, which has a $\Delta L/(\sqrt{K}R)$ factor as opposed to the $\Delta L/(KR)$ term in the lower bound. We discuss this gap further in Section C.2.

Our construction for Theorem 3.2 uses the non-convex hard instance proposed by Carmon et al. [32] and splits it across different machines to get a communication complexity lower bound. This idea has been used previously to give lower bounds in the heterogeneous setting [33, 15, 34]. We prove the result in Appendix B. From looking at Table 1, we can note that BVR-L-SGD [22] also attains a similar upper bound as our method. In Appendix C.2, we show that with deterministic oracle BVR-L-SGD also attains the min-max optimal rate. This is not surprising, knowing that several variance-reduced algorithms [25, 26, 24] are simultaneously optimal even in the sequential setting. Still, our method requires fewer and lighter variance reduction operations, which leads to better scalability from the algorithmic design perspective. In the next section, we carefully examine the difference between these methods.

## 3.1 The Perspective of Reducing Communication

So far, we have looked at convergence rates in the intermittent communication model, where $K, R$ is fixed. However, another perspective is reducing the communication complexity to the minimum possible with the minimum required oracle complexity. Both these complexities can be expressed in terms of $\epsilon$ using the convergence guarantees we showed, where we want to attain an $\epsilon$-approximate stationary point. This view is often more useful when communication rounds comprise the bulk of the required physical time. This scenario is common in FL, where devices become available intermittently, which delays the synchronous updates. With this in mind, we summarize the communication and oracle complexities attained by both our method and BVR-L-SGD [22] in Figure 1 when optimizing with stochastic oracles. Notice that the figure has three different regimes based on the relative scaling of $\tau$ versus $\epsilon$. We focus on the green regime characterized by $\epsilon^{1/2} \in (0, \tau\sigma/(LM)]$. This regime is of most practical interest in deep

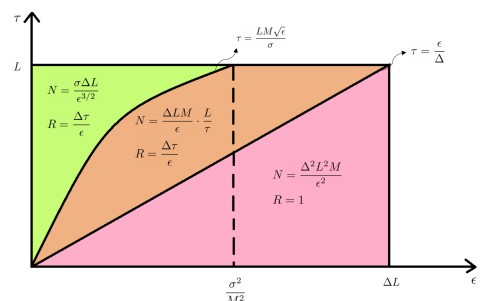

Figure 1: Illustration of the best communication complexity $R$ and oracle complexity $N$ that our method can obtain for different $\epsilon$ and $\tau$. Green regime: Our method can obtain the optimal communication and oracle complexity. Orange regime: Our method can obtain the optimal communication using a larger oracle complexity. Red regime: Our method only needs one round of communication using a larger oracle complexity.

learning, where modern over-parameterized models often drive $\epsilon$ to really small values. And when $\epsilon$ is small enough, this regime covers a wide range of values of $\tau$.[3]

In the green regime, both CE-LSGD and BVR-L-SGD require $K = \sigma L/(\tau M \epsilon^{1/2})$ local steps to achieve the optimal communication and oracle complexities. However, BVR-L-SGD requires multiple heavy-batch stochastic gradient computations on each machine with batch size $b_{max}$. In particular, for BVR-L-SGD, we have $\rho_{\text{BVR}} = b_{max}/K = \sigma\tau/(L\epsilon^{1/2})$, which suggests that for $S = L\Delta/\sigma\sqrt{\epsilon}$ communication rounds, it requires each machine to compute $\rho_{\text{BVR}}$ times heavier batch stochastic gradients compared to the other communication rounds. As for CE-LSGD, we have

---

3   We talk about the other regimes while giving the full statement of Theorem 3.1 in Appendix C

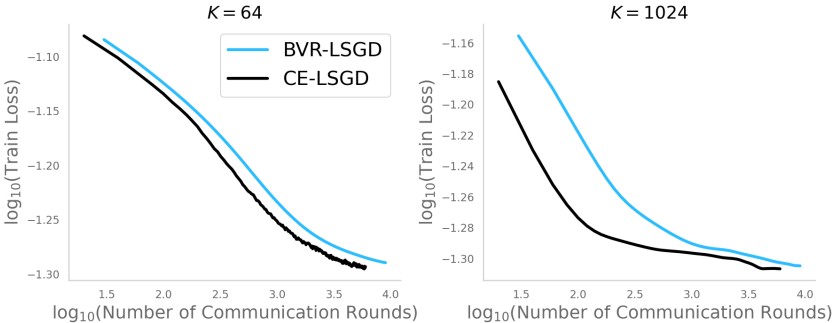

Figure 2: Training loss of CE-LSGD and BVR-L-SGD on CIFAR-10 data-set versus the number of communication rounds in the intermittent communication setting with different local-updates $K$. We use $M = 10$ machines, and synthetically generate heterogeneous data-sets (see Section 4) with $q = 0.1$. All oracle queries use a mini-batch of size $b = 16$, i.e., each machine has $Kb$ oracle queries between two communication rounds. We note that our method has a faster convergence in all the settings, which highlights its communication efficiency. Fixed step-sizes $\eta$ for both the methods were tuned in $\{0.001, 0.005, 0.01, 0.05, 0.1, 0.5\}$ (to obtain best loss) following [22], our method set the momentum $\beta = 0.3$, $b_{max}^{our} = K$, while $b_{max}^{BVR} = 5000$ according to [22].

$b_0 = \sigma^3/(L\Delta M\epsilon^{1/2})$, which gives us $\rho_{\mathrm{our}} = b_0/K = \sigma^2\tau/(L^2\Delta)$. This suggests that our method only requires each machine to compute $\rho_{\mathrm{our}}$ times larger batch stochastic gradient, and that too only once. Furthermore, $\rho_{\mathrm{our}}/\rho_{\mathrm{BVR}} = \sigma\epsilon^{1/2}/(L\Delta) \leq 1$. Thus, the size of our large batch gradient is also smaller than the one for BVR-L-SGD, and **our method has fewer and lighter heavy-batch operations**.

Suppose one implements both these methods in the intermittent communication model, i.e., by breaking the large batch computation across multiple rounds, with local budget $K = \sigma L/(\tau M\epsilon^{1/2})$. In that case, the effective communication complexity of both methods is $\Delta\tau/\epsilon$, and this subtle difference gets washed away. However, in Figure 2, we show that this equivalence up to numerical constants doesn't hold in practice, where our method converges faster than BVR-L-SGD. In Table 2, we summarize the communication and oracle complexities attained by different algorithms in the green regime.

## 3.2 The Partial Participation Setting

In settings such as cross-device federated learning [17], there are often millions of clients (think of android mobile users), and it is not feasible to consider training on all of the clients synchronously. It is more natural to consider a partial sampling of clients for each communication round. More formally, we can re-state our distributed optimization problem as follows:

$$\min_{x\in\mathbb{R}^d} F(x) := \mathbb{E}_{m\sim\mathcal{P}}\left[F_m(x)\right], \tag{3.1}$$

where $\mathcal{P}$ is a probability distribution on the clients, we assume at each communication round, we can sample $M$ clients independently from $\mathcal{P}$. We also need to modify the IC setting: during each communication round, $S_r \sim \mathcal{P}^m$ clients participate, and each queries their oracle $K$ times. This setting has also been considered in Karimireddy et al. [21]. We consider the problem classes $\mathcal{F}_{\mathcal{P}}^1(L, \Delta, \zeta)$ and $\mathcal{F}_{\mathcal{P}}^2(L, \Delta, \tau)$ that are natural generalizations of $\mathcal{F}_M^1(L, \Delta, \zeta)$ and $\mathcal{F}_M^2(L, \Delta, \tau)$ to the partial participation setting as follows, formally defined in Appendix A.

We adapt Algorithm 1 to the partial participation setting in Algorithm 2 by communicating with only $M$ clients at each round and using $M_0$ clients for the first round to initialize the variance reduction term. We prove the following guarantee for Algorithm 2.

**Theorem 3.3.** *Suppose for all $m$ in support of $\mathcal{P}$, $F_m \in \mathcal{F}_{\mathcal{P}}^1(L, \Delta, \zeta) \cap \mathcal{F}_{\mathcal{P}}^2(L, \Delta, \tau)$ then,*

*(a) if each client $m$ has a stochastic oracle $\mathcal{O}_{F_m}^{2,L,\sigma}$, and assuming that $\frac{\Delta\tau}{R} + \frac{\Delta L}{\sqrt{K}R} \preceq \frac{\sigma^2}{\sqrt{MK}} + \frac{\zeta^2}{\sqrt{M}}$,*

*the output $\widetilde{x}$ of Algorithm 2 using $b_0 = K$, $M_0 = MR$, $\beta = \max\left\{\frac{1}{R}, \left(\frac{\Delta(\tau+L/\sqrt{K})\sqrt{M}}{R(\sigma^2/K+\zeta^2)}\right)^{2/3}\right\}$,*

*and* $\eta = \min\left\{\frac{1}{L}, \frac{1}{K\tau}, \frac{1}{\sqrt{KL}}, \frac{\sqrt{\beta M}}{\sqrt{KL}}, \frac{\sqrt{\beta M}}{\tau K}\right\}$ *satisfies*

$$\mathbb{E}\|\nabla F(\widetilde{x})\|^2 \preceq \frac{\Delta\tau}{R} + \frac{\Delta L}{\sqrt{K}R} + \frac{\sigma^2}{MKR} + \left(\frac{\sigma\Delta L}{MKR}\right)^{2/3} + \frac{\zeta^2}{MR} + \left(\frac{\zeta\Delta\tau}{MR}\right)^{2/3} + \left(\frac{\Delta(\sigma\tau + L\zeta)}{M\sqrt{K}R}\right)^{2/3};$$

*(b) if each client $m$ has a deterministic oracle $\mathcal{O}_{F_m}^{2,L,0}$, and assuming that $\frac{\Delta\tau}{R} \preceq \frac{\zeta^2}{\sqrt{M}}$, then the output $\widetilde{x}$ of Algorithm 2 using $M_0 = MR$, $\beta = \max\left\{\frac{1}{R}, \left(\frac{\Delta\tau\sqrt{M}}{\zeta^2 R}\right)^{2/3}\right\}$, and $\eta = \min\left\{\frac{1}{L}, \frac{1}{K\tau}, \frac{\sqrt{\beta M}}{\tau K}\right\}$ satisfies*

$$\mathbb{E}\|\nabla F(\widetilde{x})\|^2 \preceq \frac{\Delta\tau}{R} + \frac{\Delta L}{KR} + \frac{\zeta^2}{MR} + \left(\frac{\zeta\Delta\tau}{MR}\right)^{2/3}.$$

In Tables 1 and 2, we show that with an exact oracle (i.e., $\sigma = 0$), CE-LGD attains a strictly faster convergence rate than the best-known algorithm MIMEMVR [21] that also uses an exact oracle. More specifically, CE-LGD's communication complexity $\zeta\Delta\tau/M\epsilon^{3/2}$, improves over the communication complexity of $\zeta\Delta\tau/\sqrt{M}\epsilon^{3/2}$ for MIME-MVR. We can also recover the guarantee for MB-STORM in the partial participation setting, noting that it is a special case of CE-LSGD (see Appendix C.1). As far as we know, this guarantee isn't known in the literature but straightforwardly follows from our analysis. Furthermore, we prove the following lower bounds showing that the convergence rates of CE-LSGD are *almost optimal*.

**Theorem 3.4.** *For all $L, \sigma, \tau, \Delta, \zeta \geq 0$, $\tau \leq 2L$, $\zeta \leq \sqrt{\Delta L}$, every algorithm $A \in \mathcal{A}_{zr}$ optimizing a problem in $\mathcal{F}_{\mathcal{P}}^1(L, \Delta, \zeta) \cup \mathcal{F}_{\mathcal{P}}^2(L, \Delta, \tau)$ with $K > 0$ intermittent accesses to two-point first-order oracles $\{\mathcal{O}_{F_m}^{2,L,\sigma}\}_{m \in support(\mathcal{P})}$ on all the machines outputs $x_R^A$ after $R \succeq 1$ rounds such that*

$$\mathbb{E}\left[\|\nabla F(x_R^A)\|^2\right] \succeq \min\left\{\frac{\Delta\tau}{R}, \frac{\zeta^2}{R}\right\} + \frac{\Delta L}{KR} + \frac{\sigma^2}{MKR} + \left(\frac{\sigma\Delta L}{MKR}\right)^{2/3} + \frac{\zeta^2}{MR} + \left(\frac{\zeta\Delta L}{MKR}\right)^{2/3}.$$

According to Theorem 3.3 and Theorem 3.4, in the deterministic setting (i.e., $\sigma = 0$), the only gap between the rate for CE-LGD and the lower bound is in the last term of CE-LGD's upper bound, i.e., the blue term in Table 1. We **conjecture that CE-LGD is optimal in the partial participation setting, and our lower bound can be improved**. This would also imply a gap between the optimal communication complexity of the full and partial participation settings ($\mathcal{O}(1/\epsilon)$ v/s $\mathcal{O}(1/\epsilon^{3/2})$, see Table 2). All of the known results with our partial participation setting [21] attain at best order $1/\epsilon^{3/2}$ communication complexity, which is consistent with our conjecture. More discussions about the gaps in this setting can be found in Appendix D.1.

## 4  Simulations

We evaluate the performance of our method by optimizing a two-layer fully connected network for multi-class classification on the CIFAR-10 [35] data-set. Since we are in the heterogeneous setting, we need to artificially generate a data-set. We follow the same data processing procedure as in [22]. We first make sure that all the ten classes in CIFAR-10 have the same number of samples (roughly around 5000), and assign $q \times 100\%$ of class $m$'s samples to client $m \in [10]$ where $q$ is chosen from $\{0.1, 0.35, 0.6, 0.85\}$. For each class $m$, we evenly split the remaining $(1 - q) \times 100\%$ samples to the other 9 clients except client $m$. Thus, $q$ controls the heterogeneity of our data-set, with small $q$ corresponding to small heterogeneity.

We perform two different experiments. In the first experiment, we directly compare our method, i.e., CE-LSGD, with BVR-L-SGD in the intermittent communication setting (see Figure 2). We observe that while both the methods converge to a similar quality of solution eventually, our method is more communication efficient. In the second experiment, we compare our method with BVR-L-SGD [22] as well as FEDAVG [16], SCAFFOLD [18], MB-SARAH [24] and MB-SGD [5] for the same number of updates/iterations. The last two methods are centralized baselines, and we use the local computation to compute a mini-batch stochastic gradient. We again observe that CE-LSGD and BVR-L-SGD have comparable performance which is better than all the other methods.

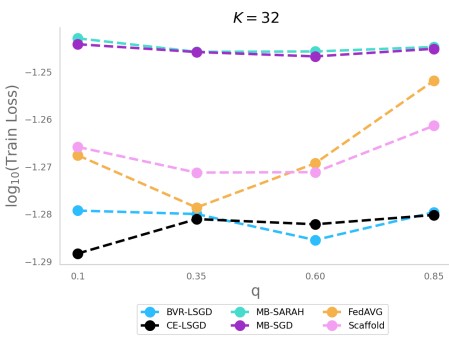

Figure 3: Comparing CE-LSGD to centralized and local-update methods, for fixed $K = 32$ and varying heterogeneity controlled by $q$ on CIFAR-10 [35] data-set. Like Figure 2 we use mini-batch size $b = 16$ for each oracle query. Thus each method makes $Kb$ oracle queries every round per machine. All the methods for different $q$ are tuned separately, following a similar hyper-parameter search as in Figure 2.

## 5 Discussion and Open Problems

In this paper, we provide a new communication-efficient local update algorithm CE-LSGD and analyze it in the full and partial client participation settings with intermittent communication. In the deterministic setting, i.e., with access to exact oracles, our algorithm is optimal for the full participation setting and almost optimal for the partial participation setting. Moreover, when equipped with stochastic oracles, our algorithm attains the best-known convergence guarantees to our knowledge in both participation models. Our lower bound results provide a much-needed baseline to measure algorithmic developments in non-convex distributed optimization and help us characterize CE-LGD's optimality.

In Appendix E, we provide an extension of CE-LSGD which uses a stochastic Hessian vector product oracle [12, 36] instead of a multi-point oracle, and recovers similar optimal communication complexity. This is relevant for memory-constrained online settings where it might not be feasible to preserve several copies of a model on the client device for making simultaneous queries for variance reduction algorithms.

Our work leaves several open questions. We believe our lower bound is loose in the deterministic partial participation setting. We expect a $\zeta\Delta\tau/M\epsilon^{3/2}$ term in the lower bound, just like our upper bound in Theorem 3.3 (c.f., the blue terms in Tables 1 and 2). Thus, we conjecture that there is a gap between the optimal communication complexities in the full and partial participation settings, order $1/\epsilon$ versus $1/\epsilon^{3/2}$. We hope to improve our lower bound in the future work.

We expect that CE-LSGD should attain the min-max optimal rate in the stochastic full participation setting. There is a $1/\sqrt{K}$ gap in our optimization term for both participation models, which vanishes in the deterministic setting (see Table 1). As discussed in Section C.2, it is unclear to us how to remove this gap.

There are several gaps w.r.t. the lower bounds in the stochastic partial participation setting (c.f., the blue, green, and red terms in Table 2). We believe some of these can be alleviated by improving the deterministic lower bound, but others seem to imply that our analysis is loose. As we discussed before, one indication that our upper bound is loose is the gap in the rate we obtain for MB-STORM by adapting our analysis for Theorem 3.3 (c.f., the red term in Table 1, section D.1).

**Acknowledgements**

We thank the anonymous reviewers who helped us improve the writing of the paper. We would also like to thank Mladen Kolar, Boxiang Lyu, Boxin Zhao, and Sebastian Stich for several useful discussions. This research was supported in part by NSF-BSF award 1718970, NSF TRIPOD IDEAL award and the NSF-Simons funded Collaboration on the Theoretical Foundations of Deep Learning.

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
