# OpenReview forum: "Towards Optimal Communication Complexity in Distributed Non-Convex Optimization"
_NeurIPS.cc/2022/Conference — NeurIPS 2022 Accept_

### Official Review · Reviewer_ssxz · 2022-06-25

**Rating:** 6
**Confidence:** 4
**Soundness:** 3 good
**Presentation:** 3 good
**Contribution:** 3 good

**Summary:**

The paper considers the nonconvex optimization problem in the centralized distributed setting and in the partial participation setting. The paper provides a new method CE-LSGD that improves previous methods by performing lighter batch stochastic gradient computations. Moreover, the paper provides the lower bounds under two heterogeneous assumptions.

**Questions:**

Can the authors get more details on why the oracle complexity of CE-LSGD is suboptimal in the partial participation setting? It is not clear how they obtained $N = O(\frac{1}{\epsilon^2}).$ Moreover, how do the authors calculate the oracle complexity of Mini-batch STORM in the partial participation setting?

**Limitations:**

,

**Strengths And Weaknesses:**

1. The first contributions **(C.1)** of the paper are the lower bounds. For simplicity, I consider Definition 2 only. Under this definition, in Theorem B.3, the authors show that $E ||\nabla f||^2 \geq \frac{\Delta \tau}{R} + ...,$ where $\tau$ is the parameters from Definition 2, while the centralized algorithm have the lower bound $E ||\nabla f||^2 \geq \frac{\Delta L}{R} + ....,$ where $L$ is Lipschitz constant such that $L \geq \tau$. I believe that these results have the following weaknesses:

    1. **Theorem B.1 (or Theorem B.3) doesn't show that $\tau$ is significantly smaller than L.** It is not clear from the example in the lower bound theorem. It means that $\tau$ and $L$ can be equal, and the author reproved the previous results. In Theorem B.3, the authors don't show that $L$ is a **tight** Lipschitz constant. In their example, the **tight** Lipschitz constant can be equal to $\tau,$ then, obviously, every constant $L \geq \tau$ is also a Lipschitz constant.
    2. The author copy-pasted the proof from Lower Bounds for Finding Stationary Points I Carmon et al., and in the proofs substituted $L$ with $\tau.$ Do not misunderstand me; there is nothing wrong with reusing (copy-pasting) the previous results and making tiny changes, but only if it leads to new results.

2. The other contributions are related to the CE-LSGD algorithm. At its core, the method is STORM (MVR) method from Momentum-based variance reduction in non-convex SGD Cutkosky el al (see also An Optimal Hybrid Variance-Reduced Algorithm for Stochastic Composite Nonconvex Optimization Liu et al.) that is extended to distributed optimization setting. As the author noted in line 117, **CE-LSGD is a mini-batch STORM, where the mini-batches are formed from the mini-batches calculated on clients**. And mini-batch STORM gets the same convergence rate as CE-LSGD. I would admit that the analysis of STORM in the distributed setting is an impactful contribution if it was not analyzed previously in the following papers:
    1. **STEM: A Stochastic Two-Sided Momentum Algorithm Achieving Near-Optimal Sample and Communication Complexities for Federated Learning Khanduri et al.** This method does not have an issue with large batch sizes, but it gets slightly worse rates due to adaptivity.
    2. **Distributed stochastic non-convex optimization: Momentum-based variance reduction. Khanduri et al.** This method does not have an issue with large batch sizes, but it gets slightly worse rates due to adaptivity.
    3. **DASHA: Distributed Nonconvex Optimization with Communication Compression, Optimal Oracle Complexity, and No Client Synchronization Tyurin et al.** This method does not have an issue with large batch sizes and gets the same rate as CE-LSGD or BVR-L-SGD.

    At least 3 papers analyzed STORM in the distributed setting. All in all, CE-LSGD improved the analysis of the previous method by considering Definition 1 and Definition 2.

3. In the partial participation setting, the oracle complexity of CE-LSGD is suboptimal w.r.t. $\epsilon.$ (see Table 2) and equals to $O(\frac{1}{\epsilon^2})$. For this method and Mini-batch STORM I would expect to get $O(\frac{1}{\epsilon^{3/2}})$ as in the full participation setting. (see Questions below)

I think that the authors are on the right way, and it is possible to improve the lower bounds by considering Definition 2 and Definition 3, but the current lower bounds theorems are not strong enough and can be misleading in the community. Moreover, a new method, CE-LSGD, doesn't provide a significant novelty that would be helpful in future research.

---

> ### Author Response · Authors · 2022-07-29
> **Clarification regarding reviewer's comments (1/2)**
>
> We thank the reviewer for their review. But we believe that the reviewer has a significant misunderstanding of our results.
>
> ### Regarding the lower bound
> >Theorem B.1 (or Theorem B.3) doesn't show that τ is significantly smaller than L. It is not clear from the example in the lower bound theorem. It means that τ and L can be equal, and the author reproved the previous results. In Theorem B.3, the authors don't show that L is a tight Lipschitz constant. In their example, the tight Lipschitz constant can be equal to τ, and then, obviously, every constant L≥τ is also a Lipschitz constant.
>
> There is a **simple way** to get $\tau$ which is always smaller than the tight Lipschitz constant. In theorem B.1 we let $F_1, F_2$ be functions from $\mathbb{R}^d\to \mathbb{R}$ but add an extra dimension in the input of $F_1^\star, F_2^\star$. Then we add the terms $\frac{L}{2}x_{d+1}^2$ to the existing $F_1^\star, F_2^\star$, i.e.,
> $$F_1^\star(x_{1:d+1}) = \frac{\tau\lambda^2}{l_1}F_1\left(\frac{x_{1:d}}{\lambda}\right) + \frac{L}{2}x_{d+1}^2,\ \text{and } F_2^\star(x_{1:d+1}) = \frac{\tau\lambda^2}{l_1}F_2\left(\frac{x_{1:d}}{\lambda}\right) + \frac{L}{2}x_{d+1}^2$$
>
> Now we can choose any $L>>\tau$, and both the functions will have a tight lipschitzness of $L$. However, when we look at the difference between the functions’ Hessians, the common term $LI_d$ will cancel out, and thus the second-order heterogeneity constant will be $\tau<<L$. The rest of the proof goes through, and we just recover the same communication complexity.
>
> We had this construction in mind but didn’t include it in the paper because we didn’t find it surprising that the tight lipschitzness constant was $\tau$. It was natural to us that the hardest instance in $\mathcal{F}_M^2(L, \tau, \Delta)$ would behave as if it were in $\mathcal{F}_M^2(\tau, \tau, \Delta)$ with or without the smoothness term. We hope this address the reviewer’s concern. We’d modify the loss in our revised version and add a remark about this. This is indeed a subtle point.
>
> >The author copy-pasted the proof from Lower Bounds for Finding Stationary Points I Carmon et al., and in the proofs substituted L with τ. Do not misunderstand me; there is nothing wrong with reusing (copy-pasting) the previous results and making tiny changes, but only if it leads to new results.
>
> This is **incorrect**. First of all, Carmon et al. consider a serial problem where there is no concept of multiple machines and hence no concept of communication. Thus their result doesn’t imply anything about the communication complexity of methods without any further arguments. It can only tell something about the iteration complexity of different algorithms. An essential tool we need on top of their hard instance is splitting the hard instance over different machines so that the zero-respecting algorithms get stuck in the co-ordinate span of the last iterate without further communication. We clearly describe this property in section B of the appendix and urge the reviewer to read these details. Even with the splitting technique, it is crucial to scale the objectives correctly to get the correct dependence in the lower bound.
>
> If, instead, the reviewer says that the hard instance is the same in our paper and Carmon et al. Then, they should note that the hard instance in their paper is essentially a modification of Nesterov’s chain function, which has been known in the optimization community **for over 25 years now**! Moreover, several other works have extended Carmon’s hard instance and Nesterov’s function to different settings. For example, see [Arjevani et al.’19](https://arxiv.org/abs/1912.02365), [Arjevani et al.’20](https://arxiv.org/abs/2006.13476), etc. What separates our paper is **the splitting technique on top of Carmon’s hard instance**, which induces the need to communicate. Our lower bound is so strong that the algorithm can have infinite oracle calls between two communication rounds (and even have access to infinite zero respecting n^th order oracle calls).
>
> We hope this clarifies that our lower bounds are not copy-pasted from any paper, especially not Carmon et al. We'd add more discussion on the lower bound proof strategy in the main body of the revised version.

---

> > ### Comment · Reviewer_ssxz · 2022-08-03
> > **Lower bound**
> >
> > Thank you for your fix. It looks like it resolves the problem. **But I encourage the authors carefully check that it does not break anything.**
> > Unfortunately, it is still not very informative. **For this particular example**, there exist algorithms **without local steps** (e.g,. MB-STORM) that will get the communication complexity $O(\frac{\tau \Delta}{\varepsilon}).$ In other words, if I run MB-STORM in this example, then I can guarantee that MB-STORM will converge after $O(\frac{\tau \Delta}{\varepsilon})$ communication rounds.
> >
> > The following question is still open: For instance, let's say that I take MB-STORM and add a second-order heterogeneity assumption. Will I improve the complexity or it is not possible even with the additional assumption?
> >
> > > This is incorrect. First of all, ...
> >
> > I don't try to attack you and understand that the progress is incremental. With your fix to the example, there is a novelty. Basically, you took the result from [Carmon et al], added a new assumption, and improved the lower bound.

---

> > > ### Author Response · Authors · 2022-08-06
> > > **We have already shown this!**
> > >
> > > >Thank you for your fix. It looks like it resolves the problem. But I encourage the authors carefully check that it does not break anything. Unfortunately, it is still not very informative.
> > >
> > > We’ve checked the details; it doesn’t break anything. We have added the construction in our revised submission.
> > >
> > > > For this particular example, there exist algorithms without local steps (e.g,. MB-STORM) that will get the…
> > >
> > > The reviewer is right that in this instance, MB-STORM sees an objective of *effective smoothness* $\tau$ up to $d$ communication rounds. And thus, the lower bound implied for MB-STORM by **this particular hard instance** is also $\frac{\Delta \tau}{\epsilon}$. But is not an issue, all it tells us, is that this instance is not the hardest for MB-STORM in $\mathcal{F}_M^2(L, \Delta, \tau)$. We clarify this below.
> > >
> > > >The following question is still open: For instance, let's say that I take MB-STORM and add a second-order heterogeneity assumption. Will I improve the complexity or it is not possible even with the additional assumption?
> > >
> > > **No, this question is not open; we have already resolved this in our paper.** In particular, copying our proof for theorem B.4 from Appendix B:
> > >
> > > “ We put the function $F$ on all the machines and endow the machines with exact oracles, i.e., $\sigma = 0$. Moreover, since this is a homogeneous problem, $\tau, \zeta=0$ for this distributed problem. Furthermore, since the oracle is queried at the same input on all the machines and returns the same fixed output, the $M$ machines can be simulated by a single machine. A single query to $O_F^{2, L, 0}$ at two different points $v, w \in \mathbb{R}^d$ is equivalent to querying the oracle $O_{F}^{1, L, 0}$ two times at $v,w$. Thus, we can implement any algorithm $A\in A_{ZR}^{cent}$ which requires $K$ total intermittent accesses to $O_{F_m}^{2, L, 0}$ for all $m\in[M]$, by instead considering a single machine with $2$ intermittent accesses to $O_F^{1, L, 0}$. Due to Carmon et al., we know that the latter problem requires at least $\Delta L/\epsilon$ oracle calls, which implies that our parallel problem requires at least $\Delta L/\epsilon$ communication rounds. This gives the first term of the lower bound $\frac{\Delta L}{R}$.”
> > >
> > > In short, even if the problem is homogeneous, i.e., we know for a fact that $\tau=0$, MB-STORM will still suffer a communication complexity of $\Delta L/\epsilon$ which is larger than what local update methods such as our method, need to suffer.
> > >
> > > We hope this resolves all the reviewer’s concerns about the lower bound, and they will further re-consider their score.

---

> > > > ### Comment · Reviewer_ssxz · 2022-08-06
> > > > **Final**
> > > >
> > > > > No, this question is not open; we have already resolved this in our paper.
> > > >
> > > > Ok, I agree with that. Basically, you mathematically formalized the following fact:
> > > > Let's say that all nodes contain the same function. Then we can find a $\varepsilon$--solution without communication between nodes, and it is not a good idea to use methods that do communication.
> > > >
> > > > This is an important remark, but its theoretical novelty and difficulty are "borderline."
> > > >
> > > > > We don’t believe this is a fair characterization of all our algorithmic results.
> > > >
> > > > Regarding partial participation, I don't like the dependences w.r.t. $\varepsilon$ in your complexities. Your paper says that CE-LSGD has the comm. complexity $O(\frac{1}{\varepsilon^{3/2}})$ and the oracle complexity $O(\frac{1}{\varepsilon^{2}}).$ Both of them are suboptimal or am I missing something? In your comment (Clarification regarding reviewer's comments (2/2), Regarding the rates in the table) you say that it is possible to get the communication complexity $O(\frac{1}{\varepsilon}),$ but I don't see in the tables or Section 3.2. Can you clarify this?

---

> > > > > ### Author Response · Authors · 2022-08-08
> > > > > **Further clarification on the centralized lower bound**
> > > > >
> > > > > ### Conclusive comments on the lower bound
> > > > >
> > > > > >Ok, I agree with that. Basically, you mathematically formalized the following fact: Let's say that all nodes contain the same function. Then we can find a $\epsilon$-solution without communication between nodes, and it is not a good idea to use methods that do communication.
> > > > >
> > > > > To be clear, this is not exactly the argument. The point of this hard instance (in theorem B.4) is **not to establish a gap between the methods that communicate and which don't**. By default, we are in the intermittent communication setting, so all methods communicate. The machines **can never know apriori**, that all of them have the same function, and they needlessly average the same gradients (when $\sigma=0$). This argument is essential in extending the serial lower bounds to the distributed setting.
> > > > >
> > > > > The reviewer's original question was whether making a $\tau$-heterogeneity assumption will provably **not help** centralized methods such as MB-STORM. The hard instance in theorem B.4 has a heterogeneity zero, which means it is the **easiest problem** in the class $F_M^2(L, \Delta, \tau)$. However, still, centralized methods incur a communication cost of $\frac{\Delta L}{\epsilon}$, which is not any better than not making any heterogeneity assumption.
> > > > >
> > > > > In other words, even the simplest problem in $F_M^2(L, \Delta, 0)$ is as hard as the most challenging problem in $F_M^2(L, \Delta, L)$ for centralized methods. On the other hand, our upper bounds show that CE-LSGD (and also other methods like MIME-MVR) have a constant (w.r.t., $\epsilon$) communication complexity in the worst case for problems in $F_M^2(L, \Delta, 0)$.
> > > > >
> > > > > >This is an important remark, but its theoretical novelty and difficulty are "borderline."
> > > > >
> > > > > We understand why the reviewer might not think this remark is difficult to make. But we don't think the argument is pedantic and that mathematically formalizing it is crucial and novel for the distributed setting. For instance, one needs to be very careful while considering **seven different terms** in Theorem 3.4, each of which requires its construction.

---

> > > > > ### Author Response · Authors · 2022-08-08
> > > > > **Upper bounds in the partial participation setting**
> > > > >
> > > > > **We'd first like to thank the reviewer for actively and patiently engaging in the discussion and providing very insightful comments.** These valuable suggestions have helped us improve our presentation and resolve the confusing points in our writing.
> > > > >
> > > > > ### Regarding the rates in the partial participation setting
> > > > >
> > > > > We are sorry about the confusion regarding CE-LSGD's communication and oracle complexities in terms of $\epsilon$ in the partial participation (PP) setting. In our revised version, we have updated Table 2 and the discussion to obtain the rates in the table (lines 703-726). We urge the reviewer to check if this resolves their concerns. In addition, the following comments are in order:
> > > > >
> > > > > 1. The confusion in our previous presentation was that there are several problem parameters $\tau, \zeta, m,$, etc., and it is unclear (without further assumptions) how they scale w.r.t. $\epsilon$. To resolve this, all rates in the revised Table 2 assume small enough $\epsilon$, and we obtain the following complexities:
> > > > >  $$R= \frac{\zeta\Delta \tau}{m\epsilon^{3/2}}, N= \frac{\zeta\Delta L}{\epsilon^{3/2}}\cdot \frac{L}{\tau} + \frac{\sigma \Delta L}{\epsilon^{3/2}}\cdot \left(1 + \frac{\sigma\tau }{\zeta L}\right).$$
> > > > >
> > > > > These rates look slightly different from the previous ones but are more intuitive for comparison because we only consider smaller values of $\epsilon$ (arguably the relevant regime). We include the full convergence guarantees in Table 1 and Table 2 is for the following regime, $$\epsilon^{1/2} \preceq \min \left(\frac{\sigma\tau}{ML}, \frac{\Delta L}{\sigma},  \frac{\Delta \tau}{\zeta}, \frac{\zeta}{m}\right).$$
> > > > >
> > > > > These conditions help us ignore the lower order terms in $1/\epsilon$. The first inequality ensures we are in the green regime described in Figure 1, and guarantees that $\frac{\Delta LM}{\epsilon} \preceq \frac{\sigma \Delta L}{\epsilon^{3/2}}$; the second inequality guarantees that $\frac{\sigma^2}{\epsilon} \preceq \frac{\sigma \Delta L}{\epsilon^{3/2}}$; the third inequality guarantees that $\frac{\zeta^2}{m\epsilon} \preceq \frac{\zeta\Delta \tau}{m\epsilon^{3/2}}$; and the fourth inequality guarantees that $\frac{\Delta L}{\epsilon} \preceq \frac{\zeta\Delta L}{m \epsilon^{3/2}}$.
> > > > >
> > > > > 2. The reviewer should note that the best communication complexity obtained by existing methods is at the order of $1/\epsilon^{3/2}$ with the oracle complexity $1/\epsilon^{3/2}$. At first glance, this seems sub-optimal because, in the full participation case, we can get communication complexity $1/\epsilon$ with these many oracle calls. But note that our lower bound for centralized methods already implies that in the partial participation setting, these methods (including MB-STORM) **must incur $1/\epsilon^{3/2}$ communication rounds**. This is because of the lower bound term $(\zeta\Delta L/mR)^{2/3}$ in Theorem B.6.
> > > > > While we cannot yet prove a $1/\epsilon^{3/2}$ lower bound for all distributed zero respecting algorithms, **we conjecture that there should be a  $\frac{\zeta\Delta\tau}{m\epsilon^{3/2}}$** term in Theorem 3.4. And thus, **there is a gap between the communication hardness of the full and partial participation settings**.
> > > > > 3. Having said all of this, our method still obtains **the best-known convergence guarantee in the partial participation setting**, which is better than previous local update methods [Mime](https://arxiv.org/abs/2008.03606), [Fed-Page](https://arxiv.org/abs/2108.04755) as well as MB-STORM. Please check tables 1 and 2 to compare these methods.
> > > > > 4. Finally, our current rates also highlight a gap between the upper and lower bounds in the centralized setting. As we remarked before, we don't know of any work analyzing MB-STORM in this setting. We believe our upper bound for CE-LSGD (and thus for MB-STORM, a particular case) is loose, which results in these gaps. We leave the task of improving this upper bound and the partial participation lower bound for future work.

---

> > > > > > ### Comment · Reviewer_ssxz · 2022-08-08
> > > > > > **Complexities**
> > > > > >
> > > > > > Thank you.
> > > > > >
> > > > > > Now, you improved the oracle complexity from $\frac{1}{\varepsilon^2}$ to $\frac{1}{\varepsilon^{3/2}}.$ By assuming that $\varepsilon$ is small enough. Looking more at the details, I noticed that you require a bounded gradient assumption (Definition 1). It was not so important in the full participation setting because there is $\min(...).$ In Table 2, CE-LGD and Lower Bound depend on $\zeta.$
> > > > > >
> > > > > > Investigation closer Theorem 3.4 (Theorem B.5), I think that I found a flaw:
> > > > > > I understand your argument that reduces the partial participation problem to **the stochastic problem**. However, you are missing one important thing, that this problem is actually **the finite sum problem,** and you have a completely different oracle. **You referred to Arjevani et al. and used their result. Unfortunately, you can not do that because the class of functions that they consider is wider.** Note that the worst-case functions that they use do not belong to the set of functions with the finite sum structure, so you can not use their result.
> > > > > >
> > > > > > I will temporarily decrease the score until I get a comment from the authors.

---

> > > > > > > ### Author Response · Authors · 2022-08-08
> > > > > > > **The definition of the Partial Participation Case**
> > > > > > >
> > > > > > > We believe there is confusion regarding the partial participation setting.
> > > > > > >
> > > > > > > > Investigation closer Theorem 3.4 (Theorem B.5), I think that I found a flaw: I understand your argument that reduces the partial participation problem to the stochastic problem. However, you are missing one important thing, that this problem is actually the finite sum problem, and you have a completely different oracle. You referred to Arjevani et al. and used their result. Unfortunately, you can not do that because the class of functions that they consider is wider. Note that the worst-case functions that they use do not belong to the set of functions with the finite sum structure, so you can not use their result.
> > > > > > >
> > > > > > > Note that the objective for the full participation setting (1.1) is a finite sum problem from the machines' perspective. If we were solving the partial participation problem on this objective, the lower bound construction in Arjevani et al. wouldn't work, as the reviewer pointed out. The issue is that the statistical estimation hard instance, a random quadratic function, won't fit into the objective (1.1), as the reviewer noted. However, **we are not solving objective (1.1) in the partial participation setting**.
> > > > > > >
> > > > > > > We are solving the objective (3.1) in the partial participation setting, a stochastic optimization problem where one can't implement an active oracle, i.e., **choose specific data points from a particular machine**. Moreover, we change our problem definitions in the partial participation setting to accommodate this; see definitions 5 and 6 in appendix A. As a result, **our lower bound arguments in theorems 3.4 and B.5 for the heterogeneity terms are correct**. We use a random quadratic function (like Arjevani et al., and several other papers before them) on each machine to get the $\frac{\zeta^2}{mR}$ term in lower bounds B.5, B.6.
> > > > > > >
> > > > > > > We understand this is a subtle point and will include a formal discussion of the difference in objectives in the partial participation setting. Due to space constraints, we couldn't fit this discussion and the new definitions 5 and 6 in the current main body.
> > > > > > >
> > > > > > > > Looking more at the details, I noticed that you require a bounded gradient assumption (Definition 1). It was not so important in the full participation setting because, in Table 2, CE-LGD and Lower Bound depend on $\zeta$.
> > > > > > >
> > > > > > > In light of the above comment, it should be clear that the dependence on $\zeta$ is **unavoidable in the partial participation setting**. Unlike the full participation lower bound, where $\zeta$ only appears through the min term, $\zeta$ appears through two additional terms $\frac{\zeta^2}{mR} + \left(\frac{\zeta\Delta L}{mKR}\right)^{2/3}$ in the partial participation setting (due to the noise in client sampling). This dependence is unavoidable and presents a real gap between the full and partial participation settings.
> > > > > > >
> > > > > > >
> > > > > > > ## **Relevance of the partial participation setting**
> > > > > > >
> > > > > > > We'd like to preemptively clarify why we believe the objective in (3.1) is important. This point goes back to the difference between **cross-silo and cross-device federated learning** ([Wang et al.](https://arxiv.org/pdf/2107.06917.pdf)).
> > > > > > >
> > > > > > > Our objective (1.1) captures the cross-silo setting, where for instance, the different devices are data-centers, or even a small number of edge devices, which can be indexed and accessed as desired. The data-set sizes on each device are much larger than the number of devices, and it makes more sense to model the data-generation process on the device instead of modeling the sampling of the devices. One practical example would be user analytics happening on 100 data centers. The user data appears online, and the server doesn't save it. But the server identity and availability persist over time. Thus, the problem on each server is a stochastic (online) optimization problem, while the global objective has a finite sum structure w.r.t. the servers.
> > > > > > >
> > > > > > > On the other hand, our objective (3.1) captures the cross-device setting, where for instance, the different devices are all the android smartphones in use. In this case, the number of devices could be much larger than the data-set size on each device. And more importantly, since there is an extremely large number of devices, think of millions if not a billion smartphones; most devices would participate at most once or twice in the collaborative optimization. In this setting, we can't hope to index devices and use data from specific devices because the devices would probably be unavailable in the future! The authors of [MIME](https://arxiv.org/abs/2008.03606) also consider this setting in objective (3.1), but our results dominate their results.
> > > > > > >
> > > > > > > Thus, both these settings are essential to distributed optimization and present their challenges.
> > > > > > >
> > > > > > > We hope this resolves the reviewer's concern, and they would consider increasing their original score. Once again, we thank the reviewer for trying to get to the bottom of our results.

---

> > > > > > > > ### Comment · Reviewer_ssxz · 2022-08-08
> > > > > > > > **Respond**
> > > > > > > >
> > > > > > > > I see. Partial participation is usually meant by the sum of functions (aka finite sum form). But you consider the partial participation as the stochastic optimization problem (aka online form) when the number of nodes is infinite (as in Mime paper). I don't have more questions about the lower bounds. Thank you!
> > > > > > > >
> > > > > > > > I will increase my score.

---

> > > > > > > > > ### Author Response · Authors · 2022-08-08
> > > > > > > > > **Thanks**
> > > > > > > > >
> > > > > > > > > We thank the reviewer again! The discussion was very beneficial and helped us significantly improve our paper.
> > > > > > > > >
> > > > > > > > > >I see. Partial participation is usually meant by the sum of functions (aka finite sum form). But you consider the partial participation as the stochastic optimization problem (aka online form) when the number of nodes is infinite (as in Mime paper). I don't have more questions about the lower bounds. Thank you!
> > > > > > > > >
> > > > > > > > > We agree that the phrase "partial participation" can mean a few different things. We'd make sure to clarify this in our revision.

---

> ### Author Response · Authors · 2022-07-29
> **Clarification regarding reviewer's comments (2/2)**
>
> ### Regarding our method
>
> >The other contributions.....CE-LSGD.
>
> First of all, our method is **neither** mini-batch-STORM (where the mini-batches are formed from the mini-batches calculated on clients) **nor** a straightforward extension of STORM [26] to the distributed setting (where both the server and clients use the same STORM gradient estimator). As we mention in line 108 and line 112, the local gradient estimator (see line 14 in Algorithm 1) of our method is instead motivated by SARAH [24, 25], and the variance reduction term used in the server (see line 9 in Algorithm 1) is inspired by STORM [26].
> Since
> 1. the local gradient estimator and the variance reduction term used in the server are **different**, and
> 2. we perform local updates instead of calculating the mini-batches at the same point,
>
> our method is **not** mini-batch STORM or a straightforward extension of STORM.
>
> In line 117, we **do not say** that our method is MB-STORM but instead claim that our method **can reduce** to MB-STORM if we do not perform local updates. There is a big difference between the two.
>
> More importantly, due to our novel algorithmic design, the convergence rate of CE-LSGD is **faster than mini-batch STORM** if the heterogeneity is small (see Table 1 for the exact convergence rate, note that $\tau$ could be much smaller than $L$ as explained in response to reviewer 2).
>
> > I would admit that the analysis of STORM in the distributed setting is an impactful contribution if it was not analyzed previously in the following papers:
>
> We thank the reviewer for mentioning the three relevant papers. However, the algorithms proposed by these papers are either a straightforward extension of STORM (STEM proposed by Khanduri et al.) or just MB-STORM (DASHA proposed by Tyurin et al. and AD-STORM proposed by Khanduri et al.). The convergence rates of these algorithms are **the same as mini-batch STORM** no matter how small the heterogeneity ($\tau$) is. In sharp contrast, the convergence rate of CE-LSGD can be **much faster than MB-STORM** if the heterogeneity ($\tau$) is small. This is **a significant improvement** offered by our method over the algorithms proposed in the three papers above.
>
> We note again that the convergence rate of **DASHA** is the same as the MB-STORM, which is **worse than** BVR-L-SGD and CE-LSGD. We will discuss these three papers in more detail in the revised version.
>
> ### Regarding the issue with large batch sizes
>
> We believe the reviewer misunderstood the convergence rate and the smallest communication complexity one can obtain. To achieve the claimed convergence rate, both STEM and AD-STORM require adaptivity, leading to a worse convergence rate than MB-STORM. DASHA (DASHA-MVR) and CE-LSGD need a slightly large batch size initially. As a result, DASHA can achieve the same convergence rate as the MB-STORM, but CE-LSGD can achieve a **faster** convergence rate than the MB-STORM. To obtain the smallest communication complexity (of the order of $O(1/\epsilon)$ instead of $O(1/\epsilon^{3/2})$, see Table 2), all these algorithms require large batch sizes, i.e., large K, for local updates. Thus all of them will suffer from large batch size-related issues (as mentioned by the reviewer).
>
> More importantly, CE-LSGD can obtain a smaller communication complexity than other methods when the heterogeneity is small. We will clarify this in the revision. Due to the above reasons, we believe that CE-LSGD is significantly novel and it would be helpful in future research.
>
> ### Regarding the rates in the table
>
> > In the partial participation setting, the oracle complexity of CE-LSGD is suboptimal w.r.t. ϵ. (see Table 2) and equals to O(1ϵ2). For this method and Mini-batch STORM I would expect to get O(1ϵ3/2) as in the full participation setting. Can the authors get more details on why the oracle complexity of CE-LSGD is suboptimal in the partial participation setting? It is not clear how they obtained N=O(1ϵ2). Moreover, how do the authors calculate the oracle complexity of Mini-batch STORM in the partial participation setting?
>
> The reason that the oracle complexity of CE-LSGD and mini-batch STORM is $O(1/\epsilon^2)$ is that we aim to achieve the smallest communication complexity (at the order of $(1/\epsilon)$, see Table 2) in the partial participation setting. If we are satisfied with $O(1/\epsilon^{3/2})$ communication complexity, the oracle complexity of CE-LSGD and mini-batch STORM can be as low as $O(1/\epsilon^{3/2})$. This is one of the critical differences between partial and full participation settings. It could be an interesting future research direction to improve the oracle complexity in the partial participation setting when we aim to achieve the smallest communication complexity.
> We apologize for the confusion about the computations of the oracle complexity in Table 2. We include the detailed calculations in the revised appendix; please check the section “Oracle complexity in Table 2” around line 672.

---

> > ### Comment · Reviewer_ssxz · 2022-08-03
> > **Method**
> >
> > I see. It seems that local steps help to reduce complexity when $\tau$ is small. I agree.
> >
> > From the view of the complexities, I don't understand what do you gain compared to BVR-L-SGD? The main motivation is "our method has fewer and lighter heavy-batch operations" (line 169). I understand that BVR-L-SGD is based on the variance technique where a method "sometimes" calculates a large batch size (see SARAH or PAGE). Your method borrowed the variance technique from STORM (aka Momentum-Based Variance Reduction in Non-Convex SGD). This was the main contribution of STORM, they provided a method that doesn't calculate large batch sizes.
> >
> > Note, that on average, BVR-L-SGD still calculates a small batch size. The "large batch problem" was solved in Momentum-Based Variance Reduction in Non-Convex SGD.
> >
> > In total,
> > 1. The authors improved the lower bound by considering additional assumptions.
> > 2. They proposed a new method that does not improve the complexity of BVR-L-SGD. However, they fixed the problem of large batch sizes by using the technique from STORM method.
> >
> > I agree to increase the score slightly.

---

> > > ### Author Response · Authors · 2022-08-06
> > > **Further clarification on our algorithmic contribution**
> > >
> > > >They proposed a new method that does not improve the complexity of BVR-L-SGD. However, they fixed the problem of large batch sizes by using the technique from STORM method.
> > >
> > > We don’t believe this is a fair characterization of all our algorithmic results. Note the following (c.f., Table 1 in our revised supplementary material):
> > > 1. We provide four sets of upper bounds, stochastic and deterministic, for full (FP) and partial participation (PP) settings. Out of these, only one setting has been considered in the paper proposing BVR-L-SGD. And out of these four settings, we show that our method is optimal in one setting and almost optimal in two others (stochastic FP and deterministic PP). We believe this significantly advances our understanding of heterogeneous non-convex optimization.
> > > 2. We don't improve upon BVR-LSGD in the stochastic FP setting, partly because there is little room to improve given our lower bound. Since no lower bounds were known before with explicit dependence on the heterogeneity parameters, it was impossible to know that BVR-L-SGD was almost optimal. This fact is evident, only in retrospect, thanks to our lower bounds, and should not be taken for granted.
> > > 3. One of our paper’s significant contributions is **establishing the convergence rate of CE-LSGD in the partial participation case**, which improves over all known rates in this setting, specially MIME-MVR (which uses exact oracles). BVR-L-SGD is only analyzed in the full participation case, and it is unclear what its convergence rate will look like in the partial participation case.
> > > 4. A byproduct of our analysis is the convergence rate of the centralized method, i.e., Mini-batch STORM, in the partial participation setting. As far as we know, the convergence rate of this critical baseline is unknown in the partial participation case.
> > >
> > > Thus our work is an important step toward understanding the partial participation setting. We hope the reviewer will reconsider their score in light of the clarification about the lower bound and our algorithmic contributions.

---

### Official Review · Reviewer_jnY7 · 2022-07-06

**Rating:** 6
**Confidence:** 3
**Soundness:** 2 fair
**Presentation:** 4 excellent
**Contribution:** 3 good

**Summary:**

The paper proposes a new algorithm for non-convex optimization -- CE-LSGD, which is optimal under certain heterogeneity assumptions. The authors show the algorithm optimality via matching upper-bound (convergence rate of CE-LSGD) and lower-bound (on the discussed problem class). The algorithm is analyzed for different regimes: full and partial client participation and full and stochastic gradients. Experiments on two-layer NNs support the theory.

**Questions:**

Questions naturally arise from weaknesses:
- Do you mean generalization losses in the theorems mentioned? If yes, then vital, in my opinion, test loss comparison experiments are missing. If not, what is the role of multi-point oracles? It seems the paper lacks clarity on this question.
- Is the case when $\tau$ is much smaller than $L$ is interesting, and why? Is it practical, and why?

**Limitations:**

The second weakness ("$\tau$ is much smaller than $L$") can be considered a limitation.

**Strengths And Weaknesses:**

Strengths:
- the paper presents a good survey of existing algorithms (table 1&2);
- a lower bound for the class of functions is derived;
- the proposed algorithm matches the lower bound, thus, proving optimality;

Weaknesses:
- As far as I understood, Theorems 2.1&3.1&3.2 establish generalization bounds (in Definition 3, when defining oracles, samples $z$ are derived from the distribution ${\cal D}$). Meanwhile, experiments depict the train losses only.
- Theorem 2.1. requires $\tau \leq L$, although the question regarding how often this condition is satisfied is not discussed. The only guaranteed inequality is $\tau \leq 2L$. The paper says that $\tau$ can be much smaller but, for example, in the case of convex functions (what is a subclass of non-convex functions) $\tau \geq \frac{M-1}{M} L$.

Typos/suggestions:
- line 59: usually, the optimization problem is ${\mathbb{E}} \|\nabla F(x)\|^2 \leq \varepsilon^2$;
- page 4: I would suggest adding the definition of class ${\cal F}(\Delta, L)$ in the environment of “definition” in the text so that the reader does not search for it in the text;
- Definition 3 presumable contains two typos: on line 73, $x$ is not defined; on line 74 $\sup_{x, y}$ is redundant (otherwise, the last sentence does not make sense);
- line 100: in ${\cal O}^{2, l, \sigma^2}_{F_m}$ $l$ must be capital
- Algorithm 1, line 3: what is $b_0$? (defined only later, in Theorem 3.1)
- Algorithm 1, line 12: $x_t \longrightarrow x_r$
- Algorithm 1: I would suggest leaving a tilde out over $\widetilde{m}_r$; the tilde makes the full of different notations (e.g. line 14) algorithm even more complicated;
- Theorem 3.1: $b_0$ is an integer, rounding in the theorem is absent;
- Theorem 3.1: $1/L$ is always larger $1/\sqrt{K}L$

---

> ### Author Response · Authors · 2022-07-29
> **Response to reviewer's comments (1/2)**
>
> We thank the reviewer for their review. We’d make sure to address all the typos and incorporate the suggestions pointed out by the reviewer.
>
> >As far as I understood, Theorems 2.1&3.1&3.2 establish generalization bounds (in Definition 3, when defining oracles, samples z are derived from the distribution D). Meanwhile, experiments depict the train losses only.
>
> The reviewer is right that our setting subsumes stochastic optimization/online setting where samples come from distributions, $D_m$ on machine $M$ so that $F_m(x) := E_{z\sim D_m}[f(x;z)]$ is the test loss on machine $m$. Thus our results can provide generalization guarantees for the average objective $F$.
>
> However, our setting also includes the ERM problem—set $D_m=Unif([n_m])$ i.e., the uniform distribution over the $n_m$ samples on the machine $m$, i.e., $$F(x) = \frac{1}{M}\sum_{m\in[M]}E_{z\sim D_m}[f(x;z)] = \frac{1}{M}\sum_{m\in[M]}\frac{1}{n_m}\sum_{j\in[n_m]}f(x; z_j).$$ Thus, all our results also apply to the finite sample setting we consider in our experiments.
>
> We have conducted preliminary test loss experiments and assure the reviewer that the curves have similar behavior. We've included a **new experiment in our supplementary material** which compares BVR-LSGD and CE-lSGD on the test loss. We'd add more such comparisons in the revised version.

---

> ### Author Response · Authors · 2022-08-02
> **Response to reviewer's comments (2/2)**
>
> >Theorem 2.1. requires τ≤L, although the question regarding how often this condition is satisfied is not discussed. The only guaranteed inequality is τ≤2L.
>
> We’d first like to clarify that all our lower and upper bounds suppress numerical constants, which means a factor of $2$ in $2\tau$ or $2L$ would get absorbed in the constants. Thus there is no difference in the rates when $\tau\leq L$ v.s when $\tau\leq 2L$. We can easily change the condition in theorem 2.1 to $\tau\leq 2L$ by dividing our hard instance by $2$. We hope the reviewer appreciates that this is a cosmetic concern and not necessary to prove any of our results. We care whether $\tau$ is the same order as $L$ or is much smaller.
>
> >Is the case when τ is much smaller than L is interesting, and why? Is it practical, and why?
>
> 1. We’d first like to clarify that our results don’t require $\tau$ to be much smaller than $L$. When $\tau\approx L$ (i.e., the only difference is in numerical constants), our rates recover the optimal rates just like centralized algorithms. Otherwise, our algorithm stays optimal and improves over centralized algorithms when $\tau$ is smaller. This adaptivity to $\tau$ is the desirable property that our paper and other papers, such as [Karimireddy et al.](https://arxiv.org/abs/2008.03606) and [Murata et al.](https://arxiv.org/pdf/2102.03198.pdf), try to attain.
> 2. Secondly, the bounded heterogeneity assumptions are relatively common in the literature; for instance, see [Khaled et al.](https://arxiv.org/abs/1909.04746), [Karimireddy et al.'20](https://arxiv.org/pdf/1910.06378.pdf), [Kolsolkova et al.](https://arxiv.org/abs/2003.10422), [Woodworth et al.](https://arxiv.org/abs/2006.04735), [Karimireddy et al.'21](https://arxiv.org/abs/2008.03606), [Glasgow et al.](https://arxiv.org/pdf/2111.03741.pdf), [Murata et al.](https://arxiv.org/pdf/2102.03198.pdf), and many other papers. Several of these papers have the same second-order heterogeneity assumption as we do; we didn’t invent it.
> 3. Regarding the practicality of the assumption, we’d like to re-state an example provided by [Murata et al.](https://arxiv.org/pdf/2102.03198.pdf), who also use the same assumption (see their section 2.2). Consider $D_m$’s to be the empirical distribution over the finite data-points on machine $m$, where each machine samples $n/M$ points, i.i.d. from some distribution $D$. Then if the loss function $f(;)$ is $L$-Lipschitz, we can use the Matrix Azuma-Hoeffding inequality to imply that with high probability, $\tau=\theta\left(\sqrt{M/n}L\right) = o(L)$. Thus in federated learning (or other distributed optimization settings), it is very reasonable to assume that $\tau<<L$.
> 4. Finally, as far as we know, the second-order heterogeneity assumption is the least restrictive assumption that has been considered in this domain. But so far, we didn’t know the optimal possible rates attainable under this assumption. Thus, even if there are more realistic heterogeneity assumptions, it is unclear how to provide analyses under them. Our result is an essential step towards that goal as we systematically understand the optimal rates under $\tau$.
>
> >The paper says that τ can be much smaller but, for example, in the case of convex functions (what is a subclass of non-convex functions) τ≥M−1ML.
>
> It is not clear to us what function the reviewer has in mind. Could they please share their calculations?
>
> The reviewer has given a high score to our presentation and contribution, and we thank them for that. We’d add the promised test loss experiments in the revised version and add clarifications suggested by the reviewer. If the reviewer’s concerns are alleviated, we urge them to reconsider their score.

---

> ### Author Response · Authors · 2022-08-08
> **Request to reconsider our revision**
>
> We want to point out that our revised version incorporates all of the reviewer's suggestions. We thank the reviewer again for pointing these out. In particular, we re-state all our results to account for the $\tau<2L$ vs. $\tau<L$ issue. And we have already included a test loss experiment in the supplementary zip file. We urge the reviewer to review our revisions and reconsider their original score.

---

> > ### Comment · Reviewer_jnY7 · 2022-08-08
> > **Thanks for your response.**
> >
> > Having read the rebuttal and checked the supplementary, I decided to raise the score.

---

> > > ### Author Response · Authors · 2022-08-08
> > > **Thanks!**
> > >
> > > We thank the reviewer again!

---

### Official Review · Reviewer_pZGN · 2022-07-11

**Rating:** 6
**Confidence:** 4
**Soundness:** 2 fair
**Presentation:** 2 fair
**Contribution:** 2 fair

**Summary:**

The authors provide a communication-efficient algorithm under centralized stochastic setting. They prove both upper bound of convergence of the algorithm and the lower bound for a class of the algorithm. Together with the upper bound and lower bound, the algorithm is min-max optimal algorithm under full participation setting and near optimal for partial participation setting.

**Questions:**

1.  Table 1 seems to be confusing because it seems that the lower bound is larger than the upper bounds.

2. [33] also claims that they give algorithms with the optimal rates, what is the difference between the lower bounds and the algorithms?

3. There are lots of definitions missing in the main text, it would be better to move those algorithm definitions and some main technical difficulties in the main text, by compressing Table 1 and Table 2.

**Limitations:**

Yes

**Strengths And Weaknesses:**

strengths:

1. The authors provide a new algorithm that meets the min-max optimal bounds.

2. For Herterogenety, the author does not only consider the first order but the second-order heterogeneity.

Weaknesses see the questions part.

---

> ### Author Response · Authors · 2022-07-28
> **Clarifications regarding reviewers questions**
>
> We thank the reviewer for their review and offer clarifications about their concerns below.
>
> > For Herterogenety, the author does not only consider the first order but the second-order heterogeneity.
>
> We want to clarify this point.
> 1. In the full-participation setting, we **don't** assume first-order heterogeneity in our analysis of CE-LSGD. We include the first order heterogeneity term in the lower bound because that makes the lower bound result stronger, i.e., even if objectives have bounded $\zeta$, they satisfy the lower bound. Several papers require the first-order assumption, which is why we include it.
> 2. On the other hand, we **do** require both assumptions in the partial participation setting, but we also establish that this is **unavoidable**. Due to partial sampling of clients, the first order term contributes to the variance of the stochastic gradient estimator for $F$. Thus, even with access to an exact oracle (i.e., $\sigma=0$) on each machine, $\zeta$ would still appear in the lower bound through the term $\left(\frac{\zeta\Delta L}{mKR}\right)^{2/3}$ (c.f., Theorem 3.4). If $\zeta$ were unbounded, one can't have a finite convergence upper bound for any algorithm.
> 3. Thus,  the dependence on $\zeta$ is unavoidable in the partial participation setting, and as we show, our results are nearly tight in the term involving $\zeta$. We will clarify this point further in the paper.
>
> >Table 1 seems to be confusing because it seems that the lower bound is larger than the upper bounds.
>
> **Our lower bounds are always lower than the upper bounds** provided in the tables. Perhaps the reviewer is looking at the deterministic (i.e., $\sigma=0$) upper bounds and comparing them to the stochastic lower bounds without setting $\sigma=0$. It would be great if the reviewer could point out the actual rates confusing them. Otherwise, it is difficult to answer the question precisely.
>
> >[33] also claims that they give algorithms with the optimal rates, what is the difference between the lower bounds and the algorithms?
> 1. The lower bound shown in [33] is incorrect, as stated in Theorem 2 in their paper. [33] claims that they offer a communication lower bound of $\frac{\Delta L M}{\epsilon}$ to reach a point $x$ such that $E[||\nabla F(x)||^2] \leq \epsilon$. However, they have an error in their scaling, and what they show is a communication lower bound of $\frac{\Delta L M}{\epsilon}$ to reach a point $x$ such that $E[||\nabla F(x)||^2] \leq \frac{\epsilon}{M^2}$. Adjusting for this scaling error, their actual communication lower bound is $\frac{\Delta L}{M\epsilon}$ which is much weaker than our tight lower bound of $\frac{\Delta L}{\epsilon}$ for centralized algorithms.
> 2. Their lower bound says nothing about problems where $\tau$ is smaller than $2L$. But our lower bounds for local-update algorithms explicitly depend on $\tau$. In the extreme case when $\tau=0$ and we have a homogeneous problem, the lower bound says we need a constant number of communication rounds. Indeed we show that CE-LSGD and BVR-LSGD need constant communication rounds in this regime (see the pink region in figure 1).
> 3. Unfortunately, the lower bound in [33] has been cited many times over the last few years. The papers citing the bound refer to the dependence in $\epsilon$ while suppressing all other terms. This is precisely why no one has noticed the error in their bound so far. To clear up this confusion, we chose not to suppress any term in our rates, and write down the explicit dependence on the number of machines, etc.
> 4. Moreover, the bound in [33] doesn't apply to all zero-respecting algorithms. The authors consider a particular form of the algorithm. Thus, our result is more general. We will include a detailed comparison against [33] and pinpoint the error in their proof.
> 5. Finally, the main advantage over their algorithm Fed-PD is that CE-LSGD has a better convergence rate, its performance improves with a lower second-order heterogeneity constant $\tau$, and when $\sigma=0$ it is optimal or nearly optimal.
>
> >There are lots of definitions missing in the main text, it would be better to move those algorithm definitions and some main technical difficulties in the main text by compressing Table 1 and Table 2.
>
> We believe it is essential to include the tables in the main text because they aid in comparison to related literature. One of our main goals in the paper is to summarize the current results in this space with explicit dependence on the relevant terms. But we understand that moving definitions and discussions to the main body can clarify some aspects. We'd try to do this the best we can in the revised version.
>
> We have tried to address all the reviewer's concerns and urge them to reconsider their score in light of the clarifications. We'd be happy to clarify further if there are more questions.

---

> > ### Comment · Reviewer_pZGN · 2022-08-03
> > **Clarifications of Question 1**
> >
> > Sorry for the confusion in the questions. In Table 1,  all of the three lower bound will have the term $\frac{\sigma^2}{MKR}$, while in the Lower Bound of Partial Participation Setting $\frac{\sigma^2}{MKR}$ occurs in Theorem 3.4 but omits in Table 1.  It is correct for the order because $\frac{\sigma^2}{MKR}$ is $\mathcal{O}((\frac{\sigma \delta L}{MKR})^{2/3})$. But adding this term in the lower bound is confusing since it seems that one more term occurs in the lower bound and not in the above upper bounds. To omit the term, it should be carefully discussed in Theorem 3.1, 3.3, and D.1 on how the term will not destroy the lower bound at the beginning of the algorithms.

---

> > > ### Author Response · Authors · 2022-08-06
> > > **Missing term in the lower and upper bounds**
> > >
> > > > Sorry for the confusion in the questions. In Table 1, all of the three lower bound will have the term σ2/MKR, while in the Lower Bound of Partial Participation Setting σ2/MKRoccurs in Theorem 3.4 but omits in Table 1.
> > >
> > > We thank the reviewer for pointing out the missing term $\frac{\sigma^2}{MKR}$ in the Lower Bounds in Table 1. We believe working towards the reviewer's suggestion has significantly improved our presentation. We have added that term in the updated Table 1 in the revised supplementary material. We are sorry about the confusion this omittance caused.
> > >
> > > >It is correct for the order because σ2/MKR is O((σδL/MKR)2/3).
> > >
> > > The reviewer is right. In our previous upper bound results, we assumed that $\frac{\sigma^2}{MKR}\leq \bigg(\frac{\sigma\Delta L}{MKR}\bigg)^{2/3}$, and thus chose specific $b_0,\beta,m_0$ such that the convergence rate matched the dominating term, i.e., $\bigg(\frac{\sigma\Delta L}{MKR}\bigg)^{2/3}$. This assumption is common, especially when $MKR$ is large. Therefore, our upper bound results did not violate the lower bound results.
> > >
> > > To exactly match the $\frac{\sigma^2}{MKR}+\bigg(\frac{\sigma\Delta L}{MKR}\bigg)^{2/3}$ term in the lower bound results, we can instead choose $\beta, b_0$ a bit differently, as highlighted in the revised theorems for both full and partial participation settings.
> > > As a result, we now have the convergence rate for CE-LSGD in the full participation setting as $$\frac{\Delta\tau}{R} + \frac{\Delta L }{\sqrt{K}R} + \frac{\sigma^2}{MKR} + \left(\frac{\sigma \Delta L}{MKR}\right)^{2/3},$$ and the convergence rate of Mini-batch STORM as $$\frac{\Delta L}{R}+\frac{\sigma^2}{MKR}+ \left(\frac{\sigma \Delta  L}{MKR}\right)^{2/3}.$$
> > >
> > > As you can see from these convergence rates, our upper bound is higher (but almost optimal) than the lower bound results in Table 1. Similarly, we’ve also updated the hyperparameter selection for the partial participation setting.
> > >
> > > > But adding this term in the lower bound is confusing since it seems that one more term occurs in the lower bound and not in the above upper bounds. To omit the term, it should be carefully discussed in Theorem 3.1, 3.3, and D.1 on how the term will not destroy the lower bound at the beginning of the algorithms.
> > >
> > > We have updated the statement of Theorems 3.1, 3.3, D.1, and Table 1 in the revised supplementary material to remove the confusion about the $\frac{\sigma^2}{MKR}$ term. We urge the reviewer to check our revised theorems and tables. All our upper and lower bounds now have the $\frac{\sigma^2}{MKR}$ term, and there is no scope of confusion regarding violating the lower bounds.
> > >
> > > The reviewer had two major concerns: the missing terms in the upper/lower bounds and the comparison to [33]. We have now resolved both these issues and revised our paper. We urge the reviewer to reconsider their score. And we'd like to thank the reviewer for pointing this out, we believe this significantly improved our presentation.

---

### Meta-Review · Area_Chair_btYW · 2022-08-26

**Recommendation:** Accept
**Confidence:** Certain

**Metareview:**

This paper studied the problem of distributed stochastic non-convex optimization with intermittent communication, and considered both the full participation setting and the partial participation setting. In particular, the paper proposed a new algorithm and showed that it can improve existing methods. The weakness is in that the lower and upper bounds in the stochastic case do not match well, but I think it is ok.

**Award:**

No

---

### Decision · Program_Chairs · 2022-09-14

Accept